# Gallic Acid: A Potent Metabolite Targeting Shikimate Kinase in *Acinetobacter baumannii*

**DOI:** 10.3390/metabo14120727

**Published:** 2024-12-23

**Authors:** Mansour S. Alturki, Abdulaziz H. Al Khzem, Mohamed S. Gomaa, Nada Tawfeeq, Marwah H. Alhamadah, Futun M. Alshehri, Raghad Alzahrani, Hanin Alghamdi, Thankhoe A. Rants’o, Khaled A. G. Ayil, Abdulaziz K. Al Mouslem, Mohammed Almaghrabi

**Affiliations:** 1Department of Pharmaceutical Chemistry, College of Clinical Pharmacy, Imam Abdulrahman Bin Faisal University, Dammam 31441, Saudi Arabia; msmmansour@iau.edu.sa (M.S.G.); nztawfeeq@iau.edu.sa (N.T.); 2College of Clinical Pharmacy, Imam Abdulrahman Bin Faisal University, Dammam 31441, Saudi Arabia; 2190003024@iau.edu.sa (M.H.A.); 2200002852@iau.edu.sa (F.M.A.); 2200004027@iau.edu.sa (R.A.); 2200005916@iau.edu.sa (H.A.); 3Department of Pharmacology and Toxicology, College of Pharmacy, University of Utah, Salt Lake City, UT 84112, USA; thankhoe.rantso@pharm.utah.edu; 4Chemistry Department, Faculty of Science, King Abdulaziz University, Jeddah 21589, Saudi Arabia; kaliabdullah@stu.kau.edu.sa; 5Department of Chemistry, Faculty of Science, Umm Al-Qura University, Makkah 21955, Saudi Arabia; 6Department of Pharmaceutical Sciences, College of Clinical Pharmacy, King Faisal University, Al Ahsa 31982, Saudi Arabia; aalmoslem@kfu.edu.sa; 7Pharmacognosy and Pharmaceutical Chemistry Department, Faculty of Pharmacy, Taibah University, Al Madinah Al Munawarah 30001, Saudi Arabia; mhmaghrabi@taibahu.edu.sa

**Keywords:** metabolites, gallic acid, shikimate kinase, *Acinetobacter baumannii*, molecular simulation

## Abstract

**Background/Objectives:** *Acinetobacter baumannii* is a highly multidrug-resistant pathogen resistant to almost all classes of antibiotics; new therapeutic strategies against this infectious agent are urgently needed. Shikimate kinase is an enzyme belonging to the shikimate pathway and has become a potential target for drug development. This work describes the search for Food and Drug Administration (FDA)-approved drugs and natural compounds, including gallic acid, that could be repurposed as selective shikimate kinase inhibitors by integrated computational and experimental approaches. **Methods:** Approaches to drug design using structure-based and ligand-based methodology, in-silico screening, molecular docking, and molecular dynamics for the study of both binding affinity and stability. Experimental Validation Determination of minimum inhibitory concentration (MIC) and minimum bactericidal concentration (MBC) on *Acinetobacter baumannii* and *Enterococcus faecalis*. **Results/Conclusions:** Among them, gallic acid, obtained from plants, proved to be the most promising compound that showed sufficient binding with shikimate kinase through computational studies. Gallic acid showed very good activity against *Acinetobacter baumannii* and *Enterococcus faecalis* in the MIC and MBC assay, respectively. Gallic acid exhibited better activity against *Acinetobacter baumannii* due to the overexpression of shikimate kinase. Gallic acid has emerged as a potential therapeutic candidate drug against *A. baumannii* infection and, therefore, as a strategy against the appearance of multidrug-resistant microorganisms. This study not only identifies a novel repurposing opportunity for gallic acid but also provides a comprehensive computational and experimental framework for accelerating antimicrobial drug discovery against multidrug-resistant pathogens.

## 1. Introduction

*Acinetobacter baumannii* (*A. baumannii*) is a Gram-negative, opportunistic pathogen that has now become very important in the etiology of nosocomial infections, particularly in ICUs. The organism is associated with high morbidity and mortality because of its resistance to multiple drugs and because it causes serious infections such as ventilator-associated pneumonia, bacteremia, trauma wounds, and urinary tract infections [1,2]. For example, mortality rates related to ventilator-associated pneumonia (VAP) caused by A. Resistance rates for baumannii range from 30% to 75%, with the mortality rates of bloodstream infections being above 50% [3,4,5]. This is increased further in the immunocompromised ones [6,7]. The COVID-19 pandemic increased the cases of carbapenem-resistant *A. baumannii* due to short supplies and increased ICU admissions [8].

Surprisingly, this pathogen has acquired an unusual capability for disseminating resistance mechanisms utilizing β-lactamase production, efflux pump activation, and target site modification, making the treatment very difficult [9,10,11,12,13]. The global emergence of multidrug-resistant (MDR) and extensively drug-resistant (XDR) strains has developed therapeutic intervention challenges, leaving very few options, including last-line antibiotics such as tigecycline and colistin [14,15]. In addition, *A. baumannii* uses virulent mechanisms such as biofilm formation and toxin secretion to enhance its pathogenicity and enable its persistence in healthcare environments, leading to protracted outbreaks [16,17,18,19,20]. Therefore, developing newer therapeutic strategies against this emerging public health threat becomes an urgent priority. The shikimate pathway is crucial in most living organisms, such as plants and microorganisms, yet it is absent in mammals (Figure 1). The terminal product of this pathway, chorismate, is a necessary precursor in the production of several essential metabolites, which include but are not confined to the aromatic amino acid folic acid and quinones [21]. The Shikimate kinase enzyme, a part of this pathway, catalyzes a phosphate transfer from ATP to the 3-hydroxyl group of shikimic acid, producing shikimate-3-phosphate (Figure 2) [22]. It contains three identifiable regions: the CORE motif refers to the conserved binding loop in the enzyme, essential for its catalytic activity and structural stability; the LID denotes the region that covers the active site and is critical for adenosine triphosphate (ATP) interaction; and a region of nucleotide monophosphate (NMP) binding responsible for interaction with shikimate [23].

Recent progress in bioinformatics and computational drug design has increased the identification rate of new therapeutic agents [24,25,26]. For example, structure-based and ligand-based drug design approaches may effectively identify specific inhibitors with less toxicity. In this regard, drug repurposing represents a cost- and time-effective strategy to find new indications for the already FDA-approved drugs [27]. Drug repurposing has been utilized to find potential inhibitors of bacterial enzymes, and penicillin-binding proteins have been included among other potential bacterial targets [28,29,30,31]. The current study focuses on utilizing in-silico strategies for identifying FDA-approved drugs, non-drug compounds, and natural metabolites as selective active inhibitors of *A. baumannii* shikimate kinase. We integrated the approaches of molecular docking and molecular dynamics simulations with Density Functional Theory (DFT) computations while studying the binding affinity and stability of the proposed compounds. The experimental validations of in-silico results were determined by assessing MICs and MBCs for the compounds against *A. baumannii* and *Enterococcus faecalis* (*E. faecalis*). Thus, the findings support these compounds’ potential as effective therapeutic agents and contribute to optimizing the bioactive compounds against *A. baumannii*.

## 2. Materials and Methods

### 2.1. Chemicals

Gallic acid, penicillin G, cefuroxime, and phosphate-buffered saline (PBS) were purchased from Sigma-Aldrich (St. Louis, MO, USA).

### 2.2. Generation of Databases and Ligand Library Preparation

FDA-approved drugs and natural metabolites library was retrieved from the ZINC15 online server (https://zinc.docking.org/, accessed on 25 August 2023), a public web-accessible database containing over 750 million purchasable compounds. 1650 compounds were downloaded and saved in the 2D Structure-Data File (SDF) format and then imported into Maestro (Schrödinger Release 2021–3:Schrödinger, LLC, New York, NY, USA, 2021). Hydrogen atoms were added to 1650 compounds using the LigPrep tool in Maestro (Schrödinger Release 2021–3:Schrödinger, LLC, New York, NY, USA, 2021) The output was then energetically minimized and optimized using the OPLS3e force field, generating 3D chemical structures. Ionization was conducted with Epik at a neutral pH of 7.0 ± 2.0, ensuring stereoisomers were preserved with specific chirality. The generation of tautomers and desalting were verified, and up to 32 stereoisomers per ligand were produced. This led to the generation of 2792 compounds. To avoid stereoisomeric duplicates of some compounds, the 2792 compounds were filtered using the Filter Duplicates tool in Maestro to exclude any stereoisomeric duplicates, resulting in 1590 unique compounds.

### 2.3. Retrieval of Shikimate Kinase Crystal Structure and Preparation for Docking Analysis

Figure 3 shows the three-dimensional X-ray crystallographic structure of Shikimate Kinase obtained from *Acinetobacter baumannii* in complex with Shikimate, protein data base (PDB) ID: 4Y0A, downloaded from (https://www.rcsb.org/, accessed on 25 August 2023). The visualization of the 3D structure was carried out using Pymol [32]. Protein Preparation Wizard in Maestro was utilized to fix missing residues and side chains and ensured proper protonation states at pH 7.0 using PROPKA (Version Release 2021–3). Loops were modeled using Prime, and bond orders were assigned. The pre-processing was extended to generating the Het state for the ligand in the protein’s active site. Polarity hydrogens were added, non-essential waters were removed, and all heavy atoms converged to a root mean standard deviation (RMSD) of 0.3 Å. The overall quality of the minimized structure was verified using a Ramachandran plot compared with the unminimized structure. The entire structure was minimized and optimized using the optimized potentials for liquid simulations 3 (OPLS3) force field and prepared for docking [33].

### 2.4. Binding Site Determination and Docking Validation

The grid generation tool of the receptor was employed in generating the docking grid, specifying the 3D coordinates of the active sites of *Acinetobacter baumannii* shikimate kinase to be (5.11, 9.81, 19.51) for (x, y, z) within a confined volume of 20 Å. This therefore created a centroid within the active site of the receptor and a grid box. Accuracy was ensured by redocking of the co-crystallized ligand, and docking poses and interactions were validated using Maestro by superimposition of structures and calculations of RMSD [34,35,36]. The receptor grid was centered on the bound ligand with a van der Waals radius scaling factor of 1.00 and partial charge cutoff of 0.25. The grid box enclosed the binding site, following all default parameters and with no constraints applied. This setup was then used to repeat and validate the docking in three different screening settings.

### 2.5. Non-Covalent Docking Screening (Semi-Rigid Docking)

The ligand was docked without any constraints using the Glide tool [37], with a vdw radius scaling factor of 0.80 and partial charge cut-off of 0.15, considering ligands’ flexibility, whereas in the protein it was rigid; all the other parameters were set to their default values. GlideScore implementation in Glide was used to predict binding affinity and rank the ligands. Each ligand was rank-ordered, selecting the lowest-energy docked pose by means of the pose rank. Further screening of the compounds was done with consideration of binding scores and the exact analysis of all types of binding interactions.

### 2.6. Induced Fit Docking (Flexible Docking)

Using induced-fit docking (IFD) tool in Maestro, each ligand undergoes an initial docking using a softened potential, van der Waals radii scaling, and flexible conformational sampling. Then, side-chain prediction for a specified distance around every ligand pose is performed [38]. During this step, minimization of residues and the ligand in every protein/ligand complex pose occurs. It finally predicts a favorable binding pose based on the induced-fit docking (IFD) score.

### 2.7. Screen Based on Shape

Schrödinger’s Shape Screening tool was utilized. Shikimate was used as the reference structure in this process. Six compounds were screened using the pharmacophore volume scoring technique-scoring each compound as an ensemble of pharmacophore features made up of aromatic groups, hydrogen bond acceptors, hydrogen bond donors, hydrophobic regions, and positively and negatively charged groups. The similarity score for the shapes, Shape Sim score, was calculated from the best number of matching features among these compounds [39].

### 2.8. Molecular Mechanics-Based Re-Scoring

The binding complexes were re-docked using molecular mechanics generalized Born surface area (MM/GBSA) to improve the accuracy of affinity predictions [40]. MM/GBSA allows for increased accuracy by permitting flexibility in both the ligand and the receptor-a crucial factor for physiological relevance [41]. Consequently, an extended MM/GBSA simulation was carried out to determine the ranking of binding affinities of the identified eight hits against the target abSK. Initial XP complexes of abSK hits and shikimate were redocked in Maestro using MM/GBSA, in which flexibility was induced by adjusting the distance between hits or shikimate and abSK to 5 Å. The simulation involved the VGSB solvation model with the OPLS3 force [42].

### 2.9. Molecular Dynamics Simulation Studies

The subsequent high-scoring complex was further evaluated with molecular dynamics (MD) in maestro. Desmond System Builder was used to prepare the simulation system; then the receptor-ligand complex was soaked into the TIP3P solvent model buffer system with 0.15M sodium chloride [43]. The volume of the solvent box was minimized under the OPLS4 force field, and the final system contained approximately 25,000 atoms.

The final system was retrieved for a 100 ns MD simulation, starting with the system relaxation using the default protocol. During the simulation, the isothermal-isobaric NPT entity at 310 K temperature and 1.103 bar pressure was adopted [44]. The other parameters, including the Coulombic cutoff distance and reversible reference system propagator algorithm integrator, were kept at their default settings. Afterward, the simulation results were analyzed using the Desmond Simulation Interaction Diagram tool [45].

### 2.10. ADMET Properties and Drug-Likeness Predictions

We have used the pkCSM web server, (http://biosig.unimelb.edu.au/pkcsm/prediction, accessed on 2 November 2023) [46] to predict descriptors for both ADMET and drug-likeness properties for the final selected potential inhibitors. Eight molecular descriptors were generated in order to characterize the ADMET properties of the potential abSK hits. Additionally, Pfizer’s rule (Lipinski rule of five) was applied to predict the physicochemical properties of the potential inhibitors, focusing on essential drug-like properties such as molecular weight, octanol-water partition coefficient (logP), hydrogen bond donors, and hydrogen bond acceptors [47].

### 2.11. Quantum Mechanics and Global Chemical Reactivity

The quantum mechanics calculations of the selected best two potential inhibitors were carried out using the Gaussian 09 package (Wallingford, CT) [48]. GaussView (6-0) program was used to interpret and visualize the global energy structure of potential inhibitors [39]. The density functional theory (DFT) was selected with a reliable theoretical function method at the B3LYP/6-311++ G(d,p) as a triple zeta basis set [40,41]. All the optimized three-dimensional structures were assessed in neutral form. The optimized structures were used to investigate the frontier molecular orbital (FMO) analysis, including the lowest unoccupied molecular orbital (LUMO), the highest occupied molecular orbital (HOMO), and the energy gap (Egap). Additionally, molecular electrostatic potential (MEP) [49,50] and the global chemical descriptors were discussed, such as electron affinity (EA), ionization potential (IP), global chemical hardness (η), global chemical softness (σ), electronegativity (χ), and electrophilicity (ω) were estimated according to Koopmans’ theory [51,52,53,54,55,56].

### 2.12. Microbial Strains and Media

The bacterial cultures used in this study included *Acinetobacter baumannii* ATCC BAA-747, which was cultured on MacConkey agar Condalab, cat:1052, and *Enterococcus faecalis* ATCC 29212, which was grown on blood agar SPLM with 5% sheep blood (SPLM; cat-1009). The cultures were incubated at 37 °C for 24 h, when the bacteria had reached appropriate growth. On the following day, bacterial suspensions were prepared to the 0.5 McFarland standard (approximately 1.8 × 10^8^ CFU/mL). Seven serial dilutions for each tested compound (Penicillin G, cefuroxime, X = Gallic acid, XI = Gallic acid & penicillin G, XII = Gallic acid & cefuroxime) were made starting with an initial concentration of 10 mg/mL. The inoculum and the serially diluted test compounds were dispensed into a microdilution plate. Each tube should contain 100 µL of PBS, and 50 µL of the compounds were transferred to the next tube in the series, and the microdilution plate was incubated. After the incubation, 150 µL of each dilution was added to 50 µL of the prepared 0.5 McFarland bacterial suspension in 96-well microplates and incubated at 37 °C for 24 h. Following the incubation period, the microdilution plate was read to determine the Minimum Inhibitory Concentration (MIC) value by assessing visibility.

Next, 10 µL from each well of the microdilution plate was withdrawn and cultured on blood agar media for *Enterococcus faecalis* while using MacConkey agar for *Acinetobacter baumannii*. Finally, incubate the agar plates were incubated, and dilutions were made in a 1:4 series (1, 1/2, 1/4, 1/8, 1/16, 1/32, 1/64, 1/128, 1/256, continuing up to 1/32,768). A 150 µL of each dilution was added to 50 µL of the prepared 0.5 McFarland bacterial suspension in 96-well microplates and were incubated at 37 °C for 24 h. After incubation, 10 µL of each well was cultured to determine the MIC (minimum inhibitory concentration) and MBC (minimum bactericidal concentration) were calculated as per the Clinical and Laboratory Institute guidelines [57].

### 2.13. Statistical Analysis of Data

Data is presented as mean ± standard deviation (SD) and statistical analysis was carried out using GraphPad Prism software 8.0.2 (2019) (San Diego, CA, USA). All the antimicrobial comparisons were analyzed by one-way analysis of variance (ANOVA). Statistically significant was taken as *p* < 0.05.

## 3. Results

### 3.1. High-Quality Protein Structure Evaluation

Energy minimization has been used to remove steric clashes of the SK enzyme structure representing high-energy conformations that would most likely result in unstable complex formations during simulations. However, this minimization can also introduce some unfavorable contacts within the structure and might affect the conformation of the enzyme. Therefore, it is very important to assess the enzyme structure before and after minimization to ensure that an energy-optimized model will be used for precise prediction in both docking and simulation studies. The Ramachandran plot of the pre-minimized SK enzyme showed 92.9% of the residues in the favored region, while for the minimized enzyme, 92.3% of residues were in the favored region. Neither in pre-minimized nor minimized structures was any residue present in the disallowed regions. Figure 4 depicts the Ramachandran plots of both states.

### 3.2. Docking Studies

The co-crystalline ligand was redocked into its target abSK using the same procedure and protocol applied for the six hits to validate docking. Subsequently, rigid-body superposition was performed using Maestro’s structure superposition tool to align the predicted lowest energy conformation of the target with its corresponding co-crystalline ligand. The classical RMSD from the co-crystalline pose was calculated for the predicted binding poses, with an RMSD < 2 Å considered an effective threshold for validating correctly posed molecules [58,59]. The results showed good binding mode superimposition, with an RMSD of 0.911 Å for shikimate, reflecting the accuracy of Glide’s pose prediction (Figure 5).

The SP docking study identified eight hits, including gallic acid, 4-aminosalicylic acid, mepiroxol, paraben, mesalazine, oxypurinol, and piracetam (Figure 6). However, 4-aminosalicylic acid and mesalazine were excluded from further analysis by the aggregation advisor tool [60]. The binding affinities of the remaining six hits were assessed against abSK. Initially, the inhibition profiles of these six hits were examined by docking them into the binding pockets of the target, investigating their binding patterns, target interactions, and binding affinities compared to the reference shikimate.

### 3.3. Computational Analysis of the Six Hits Binding to abSK

The findings presented in Table 1 offer valuable insights into the binding affinities and interaction profiles of the selected hit compounds with the target enzyme abSK. The application of induced-fit docking (IFD) was crucial in generating accurate complex structures for these hits, allowing the identification of true binders that might have been initially overlooked due to poor scores (false negatives). This was accomplished by employing multiple receptor conformations obtained through the IFD protocol rather than relying on a single rigid conformation, thereby enhancing the reliability of the screening process [37]. Gallic acid exhibited significant polar interactions and high binding affinity. Key interactions include an ionic bond between Arg-153 in the core domain and the carboxylate group of gallic acid and a second ionic interaction between Arg-74 in the shikimate binding domain (SB domain) and the carboxylate group. These ionic interactions are essential for gallic acid’s strong binding affinity and stability within the active site. Additionally, gallic acid forms two hydrogen bonds: one between the Asp-50 side-chain carboxyl (SB domain) and the hydroxyl groups at C5 and C6 of gallic acid and another between the main-chain amine of Arg-134 (Lid domain). Mepiroxol, another notable compound, exhibits similar interaction patterns. It forms salt bridges with Arg-153 and Arg-74 and two hydrogen bonds involving Asp-50 and Gly-96 with its terminal hydroxyl group. These interactions contribute to its strong binding affinity and high docking score. Paraben also shows substantial binding interactions, including a salt bridge with Arg-153 and two hydrogen bonds with Arg-74 and Arg-153. These interactions support its relatively high binding affinity and docking score. Favorable binding-based 2D and 3D docking positions interact with key residues within the binding pocket, as shown in (Figure 7 and Figure 8).

### 3.4. Shape Similarity Prediction

Molecules with similar shapes often fit in the same binding pocket and have similar biological activity. This is based on several different shape-describing descriptors mentioned earlier. This has succeeded as a virtual screening methodology in a chemical library for compounds similar to a given query molecule. According to Table 2, gallic acid has the highest score of shape similarity among the tested compounds. Structural similarity between gallic acid and shikimate is crucial for entry into interactions with the abSK enzyme. Indeed, gallic acid and shikimate share several crucial structural features responsible for these compounds’ high shape similarity scores.

### 3.5. Binding Free Energies Analysis

The MM-GBSA method has been employed for post-docking analysis to confirm the binding affinity of compounds towards SK, showing variable values of free energies for each complex. It has been observed that the MM-GBSA approach is very reliable in affinity predictions, primarily providing more accurate results compared to standard docking estimates. Furthermore, this method was employed for cross-verification of the docking results, and accordingly, the range of binding free energies as calculated via MM-GBSA is presented in Table 3. These results correspond to net binding free energies of −49.19 kcal/mol for gallic acid, −43.71 kcal/mol for mepiroxol, −41.33 kcal/mol for paraben, −34.9 kcal/mol for piracetam, −12.11 kcal/mol for oxypurinol and −8.63 kcal/mol for oteracil.

### 3.6. Molecular Dynamics Simulation of Gallic Acid Binding to the abSK Target

Based on the initial docking results, gallic acid was chosen for further analysis by MD perturbation. In the MD study, both the protein’s RMSD in complex with the gallic acid was maintained below 3.0 (Figure 9A), signaling that they formed a stable complex [17,18]. Furthermore, when gallic acid was fitted on abSK target, its RMSD was maintained below that of the target protein showing that it was kept on the binding pocket [61]. Rapid ligand fluctuations briefly occurred between 40 and 80 ns, however, these were followed by the re-establishment of equilibration (Figure 9A). Prominent interactions that contributed to the binding pose of gallic acid to 4Y03 included hydrogen bonds that were supplemented by both the water bridges and ionic bonds from the ASP50, ARG74, and ARG153 residues (Figure 9B). This directly correlated with the initial docking results (Figure 7 and Figure 8 and Table 1).

### 3.7. ADMET and Drug-Likeness

Poor pharmacokinetic properties lead to massive drug wastage and increased cost in drug development. Thus, determining Absorption, Distribution, Metabolism, Excretion, and Toxicity (ADMET) becomes crucial in drug discovery and development. The ideal candidate should be effective at its site of action and exhibit good ADMET properties at therapeutic doses. In all, six drug candidates underwent extensive pharmacokinetic assessments to guide the chemist in optimizing molecular structures with strong pharmacokinetic attributes. Molecular descriptors were utilized to get an idea about absorption mechanisms and possible administration routes, as well as prediction of bioavailability, water solubility, Caco-2 permeability, and human intestinal absorption. As illustrated in Table 4, good predictions of oral absorption were made for most compounds, but modifications are required in oteracil, gallic acid, and oxypurinol to improve the permeability in Caco-2. Also, none of them could cross the blood-brain barrier or enter the central nervous system, suggesting thereby limited distribution through these pathways.

Metabolic analysis indicates that interaction with CYP450 isoforms is not apparent for the selected candidates, hence decreasing the risk for complications associated with metabolic pathways. The excretion profile was considered efficient for both total clearance and renal OCT2 activities without substrate specificity for renal OCT2. Toxicological descriptors were generated based on pharmacodynamics studies: AMES toxicity, maximum tolerated dose, hepatotoxicity, and hERG inhibition. All the compounds, except piracetam, were predicted to be nontoxic; therefore, piracetam is a candidate for further experimental testing. Finally, drug-likeness of such inhibitors was analyzed to assess their overall ADMET profiles. This evaluation utilized the well-known Lipinski’s Rule of Five as a benchmark to calculate relevant descriptors. According to Lipinski’s guidelines, a compound is generally considered suitable for development as a drug or lead candidate if it maintains a molecular weight under 500 g/mol, possesses fewer than ten hydrogen bond acceptors and fewer than five hydrogen bond donors, and has an octanol-water partition coefficient (LogP) of five or less. Deviations from more than one of these criteria could lead to suboptimal oral absorption and permeability. Results detailed in Table 5 indicate that all examined inhibitors adhere to these drug-like criteria.

### 3.8. DFT Analysis

#### 3.8.1. Structure Optimization

Frequency of all the optimized structures without any imaginary frequencies to ensure their stability at a minimum energy level. The dihedral angle indicates the planarity of the optimized structure in its ground state. The dihedral angle is planar when the angle approaches 0° or ±180°. As shown in (Figure 10), most of the molecular geometries are nearly planar, except the C-O of the hydroxyl group in Mepiroxol, which is about 22.98°, while the para-hydroxyl group in Shikimate is approximately −167.21° due to the non-planarity of the cyclohexene ring. In addition, the bond length of gallic acid between the C-carbonyl group and para-oxygen of the hydroxyl group was 5.62 Å, while in Shikimate, it was slightly longer at 5.71 Å due to the C-C SP^3^ bond in cyclohexene longer compared with the C-C SP^2^ in the benzene ring of gallic acid.

#### 3.8.2. Frontier Molecular Orbital (FMO)

Frontier molecular orbital (FMO) is crucial to understanding electronic properties with the reactivity behavior of the chemical compounds [62]. The influence of FMO on organic compounds and their biological activity, particularly in the field of antimicrobial research, has become increasingly significant [63,64,65]. FMO investigates the energy level of the highest orbital, including electrons, and the lowest orbital empty. The HOMO energy levels (EHOMO) represent the ability of a compound to act as an electron donor. A more negative number (small value) of the EHOMO refers to the compound’s inability to donate electrons easily. The EHOMO values of the best two potential inhibitors and shikimate decreased in the following order: Mepiroxol > Gallic acid > Shikimate. (Figure 11). On the other hand, the LUMO energy levels (ELUMO) indicate that the molecule can act as an electron acceptor. A more negative number of ELUMO means a rising tendency to accept electrons. The ELUMO values decrease in the following order: Mepiroxol > Gallic acid > Shikimate. Consequently, the Shikimate and gallic acid displayed a low ability to donate electrons but a high ability to accept electrons during reactions. Also, the kinetic stability of the molecule was associated with energy gap calculation (Egap). The greater the value for Egap, the more kinetically stable the molecule. Therefore, the Egap values of Shikimate and Gallic acid are highest showing the lowest tendency towards chemical reactivity.

#### 3.8.3. Global Chemical Descriptors

Ionization energy represents the least energy needed to withdraw an electron from the valence energy level of an isolated atom or compound. On the contrary, electron affinity (EA) represents the value of energy released when one electron is added to a neutral gaseous state of an atom or compound and transformed into a negative ion. The increasing ionization potential (IP) and EA values indicate that the compound is less favorable as an electron donor and more able to act as an electron acceptor [66]. Shikimate and gallic acid have the highest values of IP and EA compared to mepiroxol. The chemical hardness (η) and softness (σ) are two essential factors that are associated with Pearson’s hard and soft acids and bases (HSAB), and they describe the reactivity of compounds [64]. Harder compounds are less reactive and have a higher difference in energy between HOMO and LUMO. At the same time, a rising value of softness indicates reactivity and lower stability. The results reveal that Mepiroxol exhibits significant softness and slight hardness, indicating higher reactivity and lower stability than Shikimate and Gallic acid. In addition, the higher electronegativity means that atoms in molecules readily attract electrons toward themselves. Calculated electronegativity (*χ*) shows that Shikimate and Gallic acid present higher values and thus present a higher tendency to receive electrons from the environment than Mepiroxol. Finally, the electrophilicity index (ώ) reflects the properties of molecules as electron receivers and thus be better electron acceptors [67]. Shikimate and Gallic acid present higher electrophilicity compared with Mepiroxol, thus suggesting that they are more efficient electron receivers (Table 6).

#### 3.8.4. Molecular Electrostatic Potential

Molecular Electrostatic Potential (MEP) describes the charge distribution of the 3D optimized structure of the selected potential inhibitor, shape, size, and orientation [52]. This is one of the techniques used for evaluating the reactivity of a molecule by predicting electrophilic and nucleophilic atoms [53]. It also provides evidence of which atoms in the molecule can interact with biological systems. The negative values of red color regions correspond to the electrophilic attacks, the blue sites (positives) represent the nucleophilic attacks, and the white parts indicate a neutral atom in a molecule. Figure 12 indicated that the oxygen atom in hydroxyl, carbonyl, and pyridine-N-oxide is the most negatively charged site in all the selected compounds. On the contrary, the hydrogen that is bonded with the oxygen atom is highly electronegative; hence, its site is more positive. The central rings of benzene or cyclohexene were almost neutral.

### 3.9. Antimicrobial Activity (MIC and MBC)

Antimicrobial activity assays showed that the minimum inhibitory concentrations for X = gallic acid, XI = gallic acid and penicillin G (1:1), and XII = gallic acid and cefuroxime (1:1), in the concentration range of 0.00488–0.00976 mg/mL against *A. baumannii*, 0.156–0.625 mg/mL against *E. faecalis*, the controls used were penicillin G and cefuroxime (Figure 13). The MBCs for X, XI, and XII were between 0.00976–0.0195 mg/mL against *A. baumannii* and 0.312 to 1.25 mg/mL against *E. faecalis*, expressed in Figure 13. Individually, the least MIC obtained was 0.00488 mg/mL for X, followed by 0.00976 mg/mL for XI and 0.00976 for XII against *A. baumannii* as shown in Figure 13A. The lowest MIC against *E. faecalis* was 0.156 mg/mL for XII, followed by 0.312 mg/mL for X and 0.625 mg/mL for XI as shown in Figure 13B.

## 4. Discussion

*A. baumannii* is considered a multidrug-resistant pathogen and is the cause of many nosocomial infections. In order to survive under such extreme conditions, to be resistant to most classes of antibiotics, it happened to be one of the main threats in hospitals. One of the highly expressed enzymes in *A. baumannii* was the shikimate kinase, the inhibition of which disrupted the shikimate pathway, hence leaving the bacterium devoid of some essential metabolites that it produces [23]. This inhibition thus interferes with its proliferation and survivability and thus forms a promising strategy for infection management without causing any damage to the host human being. Another Gram-positive bacterium causing nosocomial infection is *E. Faecalis*, especially in immunocompromised individuals. Its shikimate kinase expression is not as high as that of *A. baumannii*; still, shikimate kinase (SK) plays a crucial role in the metabolism of the bacterium [68]. Based on that, we selected the two above-mentioned bacterial strains for testing to investigate the antibacterial effect of the obtained hits that would be correlated with their difference in SK expression. However, SK inhibition still can be one step toward efficient treatment in the case of *A. baumannii* and *E.Faecalis*, at least when traditional antibiotics cannot work.

The molecular docking study was conducted to find possible SK enzyme inhibitors of *Acinetobacter baumannii* with a series of compounds ranging from FDA-approved drugs to non-drug compounds such as metabolites. Though gallic acid proved to be the most promising, several other compounds- 4-aminosalicylic acid, mesalazine, oxypurinol, pi-racetam, and paraben- showed different levels of binding affinity with varying interaction characteristics. Because of this, 4-aminosalicylic acid and mesalazine were excluded from further analyses, as the aggregation advisor tool showed that they tended to aggregate. Though oxypurinol and piracetam showed interesting binding scores, the interaction profile for the former and its docking poses were considered much less stable than gallic acid. Paraben showed good binding affinity but was deprioritized because of a limited specificity against the target and lower predicted pharmacokinetic properties.

Gallic acid is polyphenolic in nature, and so are its derivatives. Gallic acid and its derivatives are well-documented to exert antioxidant and antimicrobial activities. Since these are secondary metabolites, gallic acid and its structure-related derivatives can inhibit microbial growth and may act against drug-resistant pathogens. They have been reported to show excellent antibacterial activities against a wide range of pathogens, including multidrug-resistant strains [69]. It has been postulated that mechanisms concerning changing properties in the bacterial membrane led to permeabilization and leakage of intracellular constituents [70]. Gallic acid derivatives also showed promising activity against Staphylococcus aureus and Salmonella enterica, with less cytotoxicity on human cells [71]. Gallic acid was found to be responsible for its antimicrobial effectiveness due to its role in the plant’s defense mechanisms [69]. Recent reports showed that nanoformulation or use in combination with antibiotics increased the bioavailability of gallic acid while reducing its possible side effects [72]. Gallic acid has shown outstanding binding affinity and unique interaction patterns toward SK compared to existing antibiotics such as carbapenem and colistin. As a non-traditional drug, gallic acid targets an absent pathway in humans, minimizing off-target effects and toxicity. To this end, gallic acid exhibited promising inhibitions against both *A. baumannii* and *E. faecalis* within the range of 0.00488 to 0.00976 mg/mL and 0.156 to 0.625 mg/mL, respectively. This suggests the bacterial target influences gallic acid’s activity, likely due to differing metabolic enzyme expressions.

The current study explored gallic acid and some of its derivatives as shikimate kinase inhibitors using different computational analyses, such as molecular docking, MD simulation, DFT analysis, and ADMET profiling. Gallic acid and its derivatives have been docked into the active site of *A. baumannii* shikimate kinase, which helped us to identify key residues associated with the binding of ligands. The obtained docking scores and their interaction profiles presented that gallic acid forms stable interactions with active-site residues, such as Asp-50 and Arg-153, through hydrogen bonding and ionic interactions. These results indicate a high affinity for gallic acid binding to shikimate kinase, confirming that it is probably a potent inhibitor.

The MD simulations are considered for long periods of time to study the stability of gallic acid in the binding site of shikimate kinase. RMSD shows the stability of the ligand-receptor complex during the simulation. Low RMSD values indicate less deviation from the initial conformation, indicating stable binding interaction [73]. Herein, the RMSD for the gallic acid-shikimate kinase complex in this study remained below 3.0 Å throughout the simulation period, thus proving the stability of the binding pose.

The DFT analysis is a quantum mechanical study which was performed to study electronic properties, including HOMO and LUMO energies, describing molecular reactivity. The DFT calculation has provided rich information about the electronic stability of gallic acid. The HOMO-LUMO energy gap reflects the chemical stability of gallic acid; the higher the gap, the lesser the reactivity of the compound and the greater the stability. Further support was provided by the calculated ionization potential (IP) and electron affinity (EA) values, which proved that gallic acid had a stable electronic structure and was thus able to show sustained inhibitory potential upon binding to shikimate kinase.

ADMET predictions form a crucial part of compound optimization regarding its drug-likeness assessment and pharmacokinetics. In silico ADMET analysis of gallic acid and its derivatives is carried out. Water solubility, Caco-2 permeability, and blood-brain barrier permeability predictions for gallic acid had been conducted to assess its potential for bioavailability and distribution. Gallic acid showed good oral absorption but poor permeability to the blood-brain barrier and hence would be appropriate against infections excluding those of the CNS. The minimal interaction of the compound with cytochrome P450 enzymes indicated low susceptibility to metabolic degradation. Efficient elimination, as given by the total clearance data, further pointed toward a low-toxicity agent. Predicted toxicity risks included hepatotoxicity, hERG inhibition, and general mutagenicity. Gallic acid was predicted to be nontoxic, in tune with its status as a safe secondary metabolite commonly found in many plants and supportive of its possible therapeutic utility.

The in-vitro antimicrobial activity showed that gallic acid (X) exerted more inhibition on *A. baumannii* (MIC; 0.00488–0.00976 mg/mL) compared to *E. faecalis* (MIC; 0.156–0.625 mg/mL). This finding supports that gallic acid is likely working through the SK inhibition mechanism, given the background of their differential SK expression. The accuracy of the inhibition assay was further proved by testing combinations of gallic acid and penicillin G (XI) and gallic acid and cefuroxime (XII). In the case of *A. baumannii*, the MICs for the combination in both cases were lower than the individual antibiotics but higher than gallic acid (0.00976 mg/mL for XI and 0.00976 for XII and 0.00488 mg/mL for X). This again emphasizes the potency of gallic acid in this SK highly expressing bacteria compared to the two tested antibiotics. In *E. faecalis*, gallic acid alone (MIC; 0.312 mg/mL) showed higher activity than XI (MIC; 0.625 mg/mL) but lower activity than XII (MIC; 0.156 mg/mL). This shows that gallic acid is less active against this bacterial strain, as evidenced by its MIC and the superior activity of its combination with cefuroxime.

Further, the structural similarities of shikimate kinases from various species suggest that identified hits might potentially inhibit the enzyme in *A. baumannii* and could also demonstrate activity on other pathogenic bacteria; however, further studies will be essential in determining the effectiveness and the spectrum of such activity against other pathogens [74].

A key limitation of the present study involves the lack of confirmation in vivo regarding findings. Whereas results may demonstrate the efficiency of gallic acid and its derivatives in vitro, the translation of such findings into the living organism is uncertain. Also, the study may not account for the potential development of resistance mechanisms in *A. baumannii* since it has been proved that it acquires resistance in many ways. Further investigation is also required where gallic acid is administered in combination with other drugs with the view of making it effective at minimum development of resistance.

## 5. Conclusions

This paper thus illustrates that gallic acid and its derivatives act against *A. baumannii* and *E. faecalis*; it is more active against *A. baumannii* because it likely demonstrates a higher expression of the shikimate kinase gene. These results confirm that gallic acid has emerged as a promising compound with broad-range anti-bacterial properties. Thus, gallic acid and its derivatives have the potential to develop new antimicrobial therapies by inhibiting the shikimate pathway. Further studies are needed to elucidate their detailed mechanism and clinical usefulness to achieve maximum effectiveness.

## Figures and Tables

**Figure 1 metabolites-14-00727-f001:**
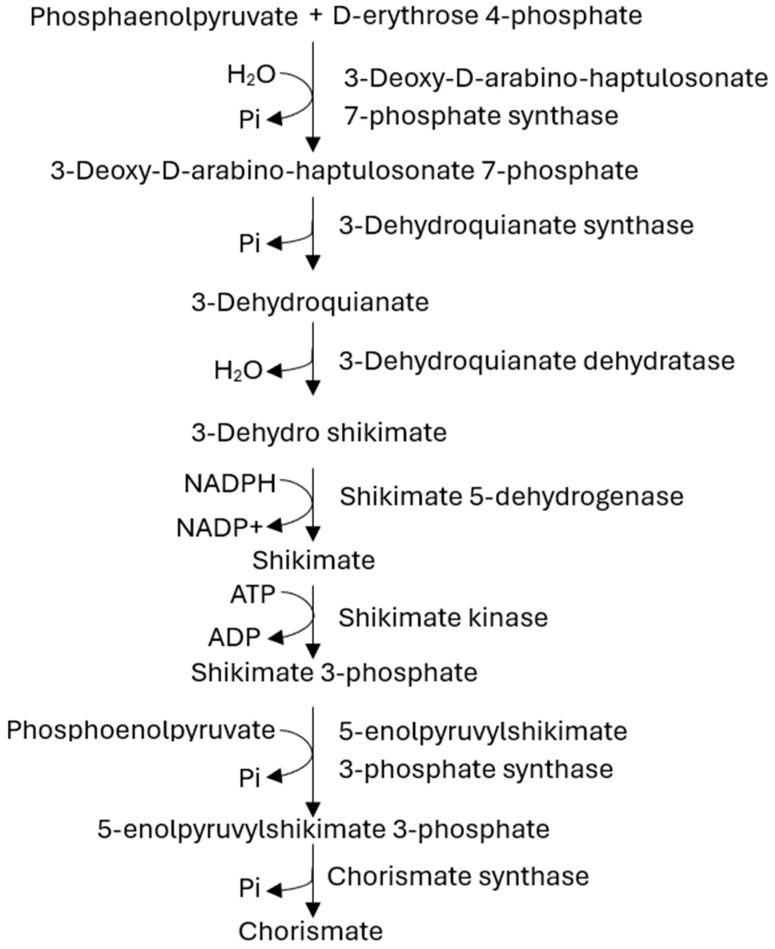
General overview of the shikimate pathway (SP), showing its importance in the synthesis of essential metabolites, including aromatic amino acids, Folic acid and quinones are crucial pathways that are absent in mammals but vital in microorganisms and plants.

**Figure 2 metabolites-14-00727-f002:**
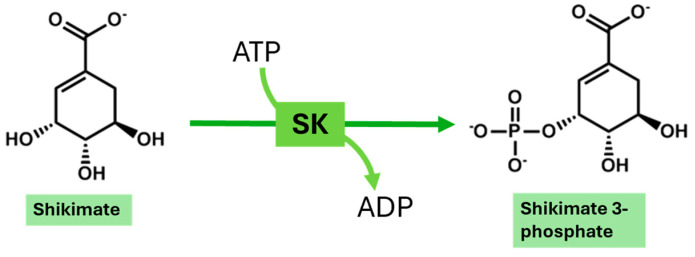
A cartoon representation of shikimate kinase (SK) in which this enzyme catalyzes the phosphorylation of shikimic acid to shikimate-3-phosphate by transferring a phosphate from ATP, essential for enzyme activity.

**Figure 3 metabolites-14-00727-f003:**
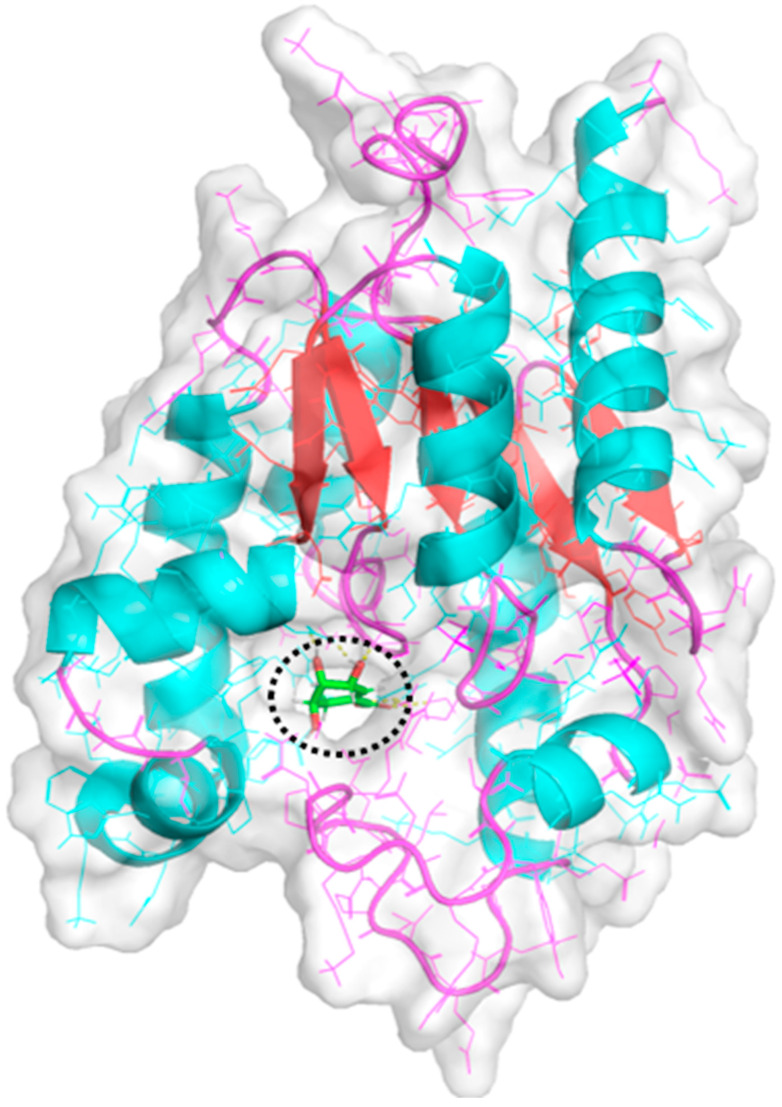
Crystal structure of *A. baumannii* shikimate kinase (abSK) bound to shikimate (SKM201) (sticks, atom color, encircled in black color). Red arrows indicate the direction of the amino acid sequence in the Shikimate Kinase enzyme.

**Figure 4 metabolites-14-00727-f004:**
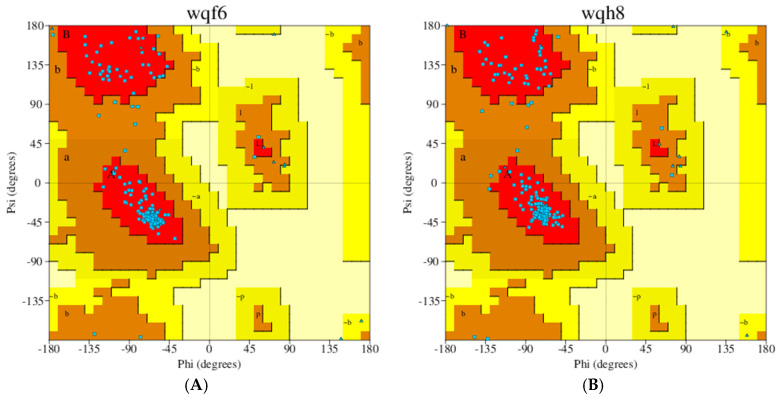
Ramachandran plot. (**A**) Energy Pre₋minimized abSK. (**B**) Energy minimized SK. Details about the coloring of the plot can be interpreted from PDBSum generated server “https://www.ebi.ac.uk/thornton-srv/databases/pdbsum/Generate.html (accessed on 25 August 2023)”.

**Figure 5 metabolites-14-00727-f005:**
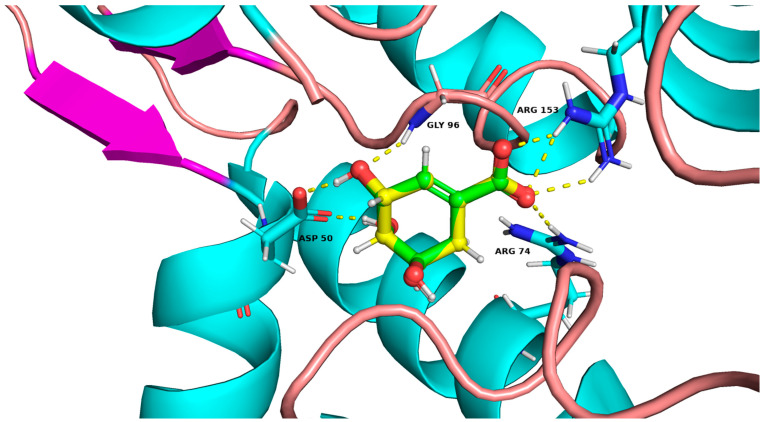
Comparison of binding poses of the co-crystal ligand (green sticks, atom color) and the redocked ligand (yellow sticks, atom color) within the abSK binding site, with an RMSD of 0.911Å. Purple arrows indicate the direction of the amino acid sequence in the Shikimate Kinase enzyme.

**Figure 6 metabolites-14-00727-f006:**
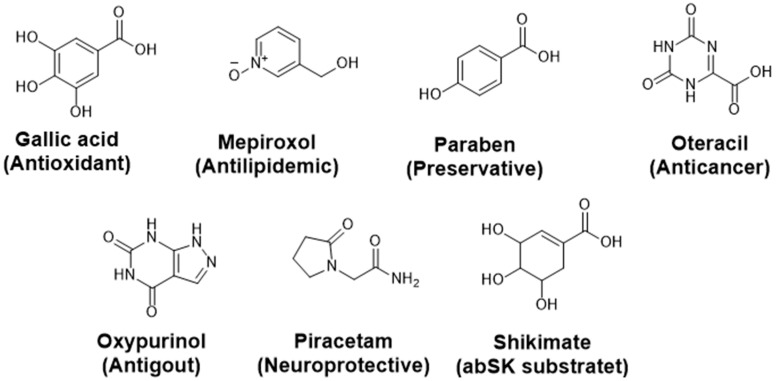
2D structures of shikimate and the six promising hits.

**Figure 7 metabolites-14-00727-f007:**
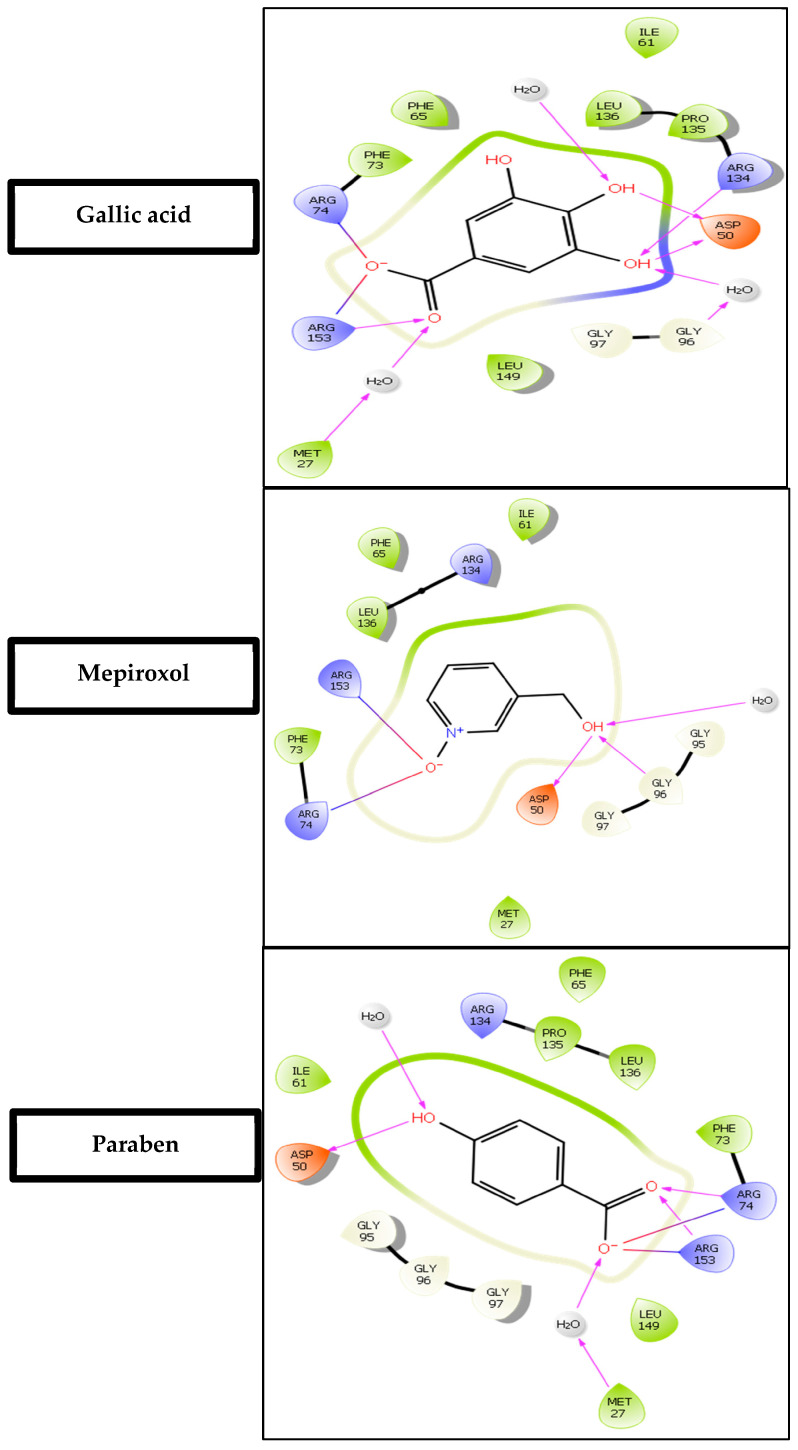
Two-dimensional ligand-protein binding interactions of abSK bounded to gallic acid, Mepiroxol, and paraben. 
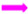
 = Hydrogen bond interactions, 
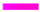
 = Ionic interactions.

**Figure 8 metabolites-14-00727-f008:**
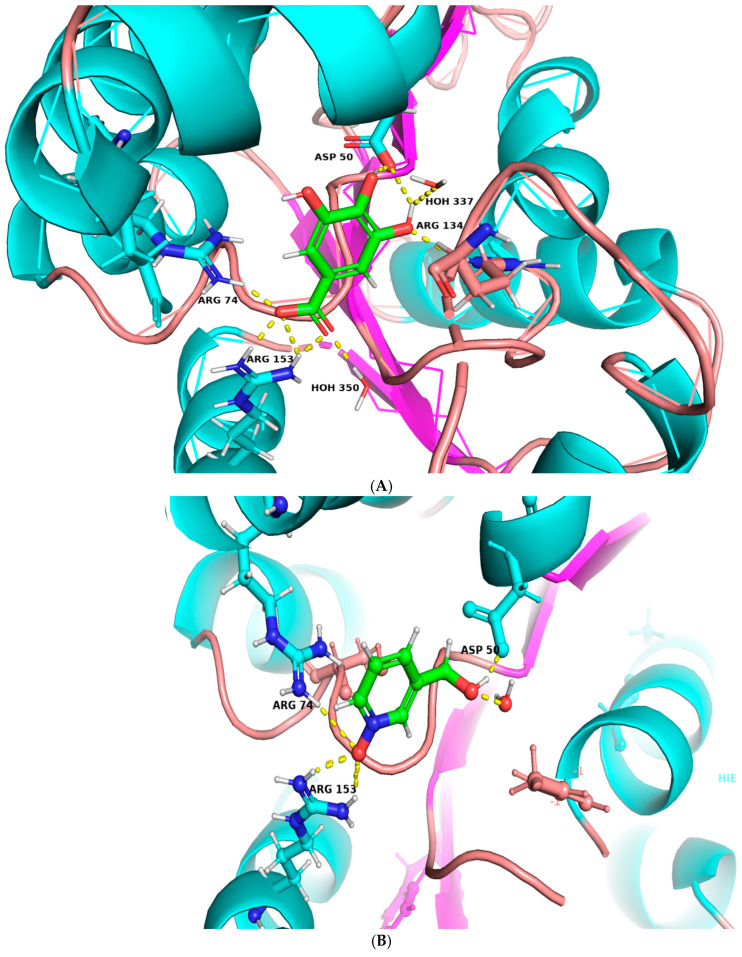
Ligand–protein binding interactions. Three-dimensional cartoon and surface representation of abSK bounded to (**A**) Gallic acid (sticks, atom color), (**B**) Mepiroxol (sticks, atom color), and (**C**) Paraben (sticks, atom color).

**Figure 9 metabolites-14-00727-f009:**
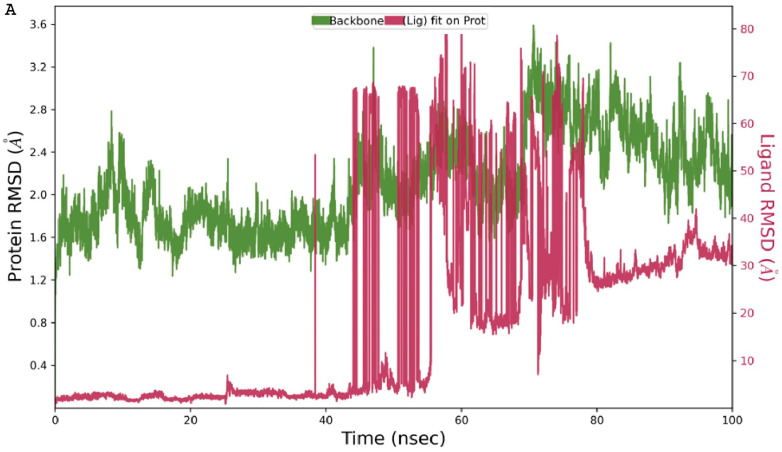
The equilibrated structural fluctuations of gallic acid and 4Y0A during the 100 ns MD simulation (**A**) (The constant interactions observed between gallic acid and 4Y0A with MET27, ASP50, ARG74, GLY96, LEU136, and ARG153 participated in the establishment of stable conformation (**B**).

**Figure 10 metabolites-14-00727-f010:**
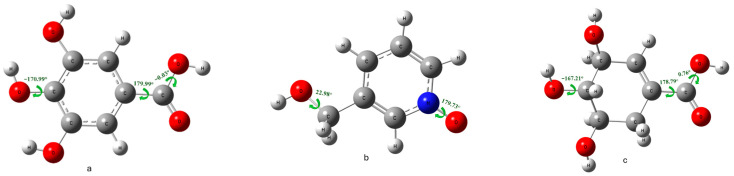
3D structures of molecular geometry optimized with dihedral angles: (**a**) Gallic acid. (**b**) Mepiroxol. (**c**) Shikimate.

**Figure 11 metabolites-14-00727-f011:**
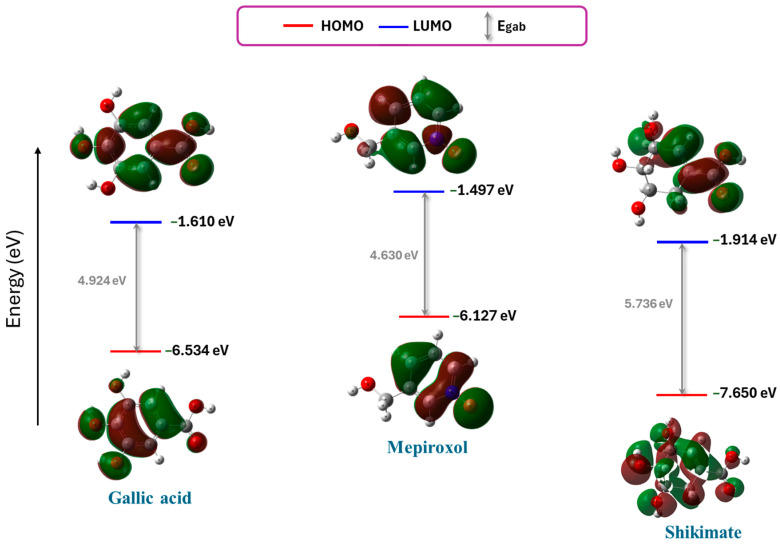
Frontier molecular orbital (FMO) including HOMO, LUMO, Egab of Gallic acid, Mepiroxol, and Shikimate.

**Figure 12 metabolites-14-00727-f012:**
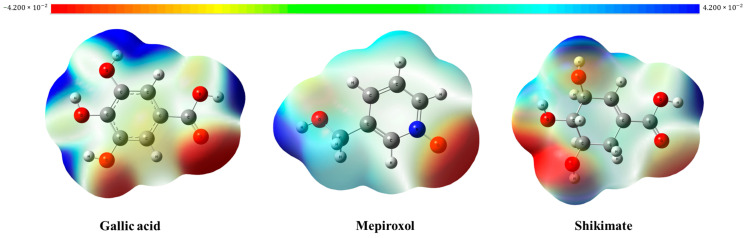
Molecular Electrostatic Potential (MEP) of Gallic acid, Mepiroxol, and Shikimate.

**Figure 13 metabolites-14-00727-f013:**
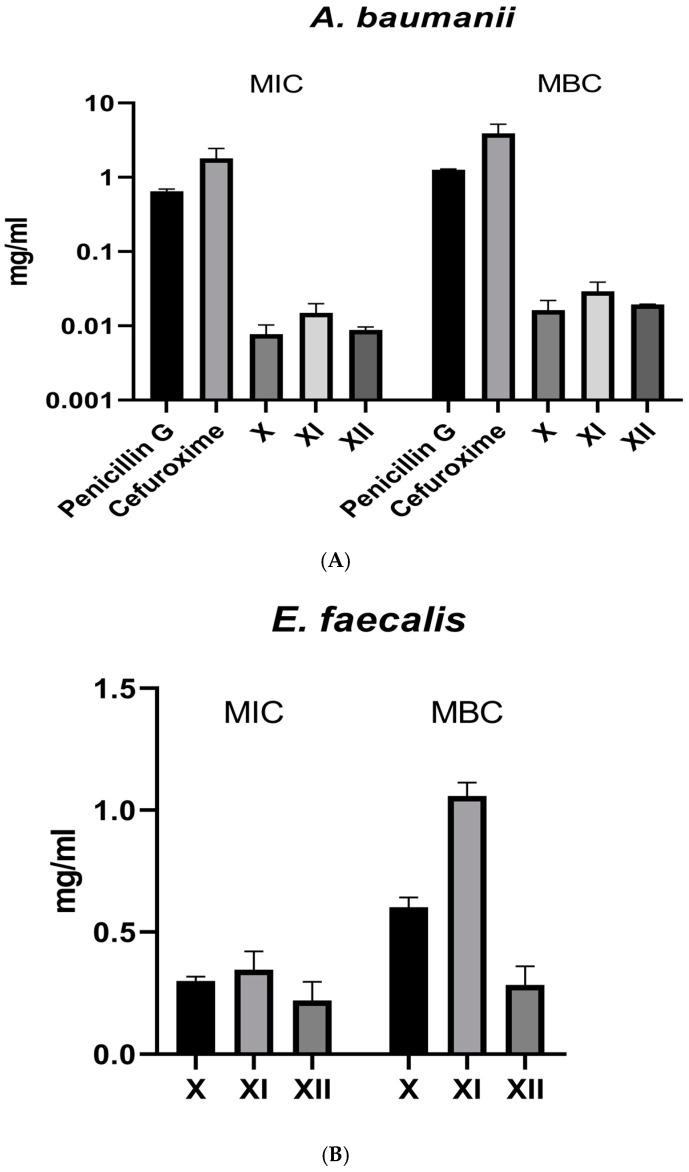
Antimicrobial results of X = gallic acid, XI = gallic acid + penicillin, XII = gallic acid + cefuroxime against *A. baumannii* (**A**) on a logarithmic scale and *E. faecalis* (**B**). Values are represented as mean ± SD, n = 3. MIC = minimum inhibitory concentration, MBC = minimum bactericidal concentration.

**Table 1 metabolites-14-00727-t001:** Docking scores and induced-fit docking of the five hits at the binding site of KHK (PDB entry: 6W0Z).

Compound/Metabolite	Glide Score Docking (Semi-Rigid) ^a^	Induced-Fit Docking (IFD) (Flexible) ^a^	Ionic Interactions (Semi-Rigid)	H-Bond Interactions (Semi-Rigid)
Gallic acid	−9.23	−9.004	ARG-74 ARG-153	ASP-50 ARG-74 ARG-134 H_2_O-350 H_2_O-361
Paraben	−6.19	−8.67	ARG-74 ARG-153	ARG-74 ARG-153 H_2_O-337 H_2_O-350
Mepiroxol	−6.22	−8.629	ARG-74 ARG-153	GLY-96 ARG-153 H_2_O-361
Oteracil	−5.98	−8.52	ARG-153	ASP-50 ARG-74 ARG-153 H_2_O-350
Oxypurinol	−4.969	−8.42	ARG-74	ASP-50 ARG-74 ARG-153 H_2_O-337 H_2_O-361
Piracetam	−4.93	−9.12		ASP-50 GLY-96 H_2_O-337 H_2_O-350
Shikimate (Control)	−9.35	−10.73	ARG-153	ASP-50 ARG-74 ARG-153 H_2_O-337 H_2_O-350 H_2_O-352 H_2_O355

^a^ Lower negative value indicates higher binding interactions within the binding pocket.

**Table 2 metabolites-14-00727-t002:** Shape similarity of the hits and control. Similarity ranges: 0.5–1 (High), ≥0.3–0.49 (Intermediate), <0.3 (Low). Cutoff score ≥ 0.4.

Compound	Shape Similarity ^a^
Gallic acid	0.596
Piracetam	0.482
Oteracil	0.447
Oxypurinol	0.426
Paraben	0.357
Mepiroxol	0.310
Shikimate (Control)	1

^a^ Values closer to 1 indicate higher shape similarity to shikimate.

**Table 3 metabolites-14-00727-t003:** MM-GBSA net binding energy of the compounds/control.

Compound	ΔG Binding ^a^	ΔG Binding Hbond	ΔG Binding vdW	ΔG Binding Solve GB
Gallic acid	−49.19	−11.60	−15.64	66.57
Mepiroxol	−43.71	−2.68	−18.34	−4.49
Paraben	−41.33	−9.82	−18.28	62.48
Piracetam	−34.93	−4.88	−17.77	26.35
Oxypurinol	−12.11	−4.23	−23.32	−49.39
Oteracil	−8.63	−10.68	−25.63	−17.76
Shikimate (Control)	−85.58	−12.08	−22.87	98.28

^a^ Lower negative value indicates higher binding affinity within the binding pocket.

**Table 4 metabolites-14-00727-t004:** ADMET profiling of the six promising drug candidates.

ADMET Parameters	Shikimate (Standard)	Oteracil	Mepiroxol	Gallic Acid	Oxypurinol	Paraben	Piracetam
Absorption							
Water solubility (log mol/L)	−0.522	−2.535	0.096	−2.56	−2.516	0.782	−0.399
Caco2 permeability (log Papp in 10–6 cm/s)	−0.23	−0.312	1.135	−0.081	−0.069	1.472	0.574
Intestinal absorption (human) (% Absorbed)	46.681	18.693	96.219	43.374	63.629	98.262	85.483
P-glycoprotein substrate (Yes/No)	No	No	Yes	No	No	Yes	No
Distribution							
BBB permeability (log BB)	−0.683	−1.005	−0.329	−1.102	−0.721	−0.039	−0.31
CNS permeability (log PS)	−3.58	−3.606	−2.857	−3.74	−3.55	−2.611	−3.098
Metabolism							
CYP2D6 substrate (Yes/No)	No	No	No	No	No	No	No
CYP3A4 substrate (Yes/No)	No	No	No	No	No	No	No
CYP1A2 inhibitior (Yes/No)	No	No	No	No	No	No	No
CYP2C19 inhibitior (Yes/No)	No	No	No	No	No	No	No
CYP2C9 inhibitior (Yes/No)	No	No	No	No	No	No	No
CYP2D6 inhibitior (Yes/No)	No	No	No	No	No	No	No
CYP3A4 inhibitior (Yes/No)	No	No	No	No	No	No	No
Excretion							
Total Clearance (log ml/min/kg)	0.688	0.897	0.867	0.518	0.733	0.597	0.613
Renal OCT2 substrate (Yes/No)	No	No	No	No	No	No	No
Toxicity							
AMES toxicity (Yes/No)	No	No	No	No	No	No	Yes
Max. tolerated dose (human) (log mg/kg/day)	0.994	1.266	0.381	0.7	1.234	1.385	1.092
hERG I inhibitor (Yes/No)	No	No	No	No	No	No	No
Hepatotoxicity (Yes/No)	No	No	No	No	No	No	Yes

**Table 5 metabolites-14-00727-t005:** Lipinski’s Druglikeness prediction of top six promising hits.

Molecule Properties	Oteracil	Mepiroxol	Gallic Acid	Oxypurinol	Paraben	Piracetam
Molecular Weight	157.085	125.127	170.121	152.112	138.123	142.157
LogP	−1.022	1.142	−0.568	−0.777	0.583	−1.884
#Rotatable Bonds	1	2	4	0	2	2
#Acceptors	6	3	4	4	3	6
#Donors	2	1	4	2	2	2
Lipinski alert	Pass	Pass	Pass	Pass	Pass	Pass

**Table 6 metabolites-14-00727-t006:** Quantum chemical descriptors of gallic acid, Mepiroxol and shikimate.

Molecule Properties	Gallic Acid	Mepiroxol	Shikimate
Ionization potential (IP) = −E_HOMO_	6.534	6.127	7.650
Electron affinity (*E**A*) = −E_LUMO_	1.610	1.497	1.914
Global hardness (η) $=\frac{IP-EA}{2}$	2.462	2.315	2.868
Softness (σ) $=\frac{1}{\eta }$	0.406	0.432	0.349
electronegativity (*χ*) = $\frac{IP+EA}{2}$	4.072	3.812	4.782
Electrophilicity index (ώ) $=\frac{\chi^{2}}{2\eta }$	3.367	3.138	3.987

## Data Availability

The original contributions presented in this study are included in the article. Further inquiries can be directed to the corresponding authors.

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
