# Peer review of "Gallic Acid: A Potent Metabolite Targeting Shikimate Kinase in *Acinetobacter baumannii"

_metabolites, 2024, doi:10.3390/metabo14120727_

Round 1

Reviewer 1 Report

Comments and Suggestions for Authors

In the manuscript “Targeting Shikimate Kinase in Acinetobacter baumannii: Gallic Acid as a Potent Inhibitory Metabolite from FDA-Approved Drug Screening”. This work describes an in-silico study identifying FDA-approved drugs that could be repurposed as selective inhibitors of A. baumannii shikimate kinase. The A. baumannii is a Gram-negative, as one of the leading causes of nosocomial infections, especially in ICUs. The study explored gallic acid and some of its derivatives as shikimate kinase inhibitors by using different computational analyses, such as molecular docking, MD simulation, DFT analysis, and ADMET profiling. They found that, the gallic acid and its derivatives have the potential to develop new antimicrobial therapies by inhibiting the shikimate pathway and contribute to the optimization of the bioactive compounds against A. baumannii. The manuscript is generally well-addressed and well-cited; however, I have some comments/suggestions.

Line 4: I suggest rewriting the title to be representative and concise

Line 27: Please rewrite abstract by the way of the journal guidelines. Please start first by Background then Objectives, not the opposite. Then, Methods, Results, and Conclusions.

Line 33: Please verify these abbreviation "MIC and MBC" as first time mentioned on the manuscript.

Line 40: The introduction is so long. Please rewrite to be more focus and specific.

Line 57: Please note that more than 10 lines discuss the % of morbidity. Please rewrite it in brief again.

Line 91: Please verify these abbreviation "CORE and LID" as first time mentioned on the manuscript.

Line 107: Please note that sometimes the text discussing the antibiotics like that one from line 105 until 107. After that, the text comes back to talk about the targeted bacteria at this work study, “A. baumannii”. That's repeated many time in introduction. Please keep more focus on the literature concerning your targeted bacteria and their previously published data and how did you used it to build up your aim of the study.

Line 126: Please rewrite the figure legend to be more informative.

Line 129: Please rewrite the figure legend to be more informative.

Line 148: Please rewrite this paragraph by mentioning the  following steps (why did you do it, how did you do, how did you use it) until you get the results of this 3D structure. It is not recommended to start by the results of 3D structure directly.

Line 162: please add the meaning/ indication of red arrows to the figure legend.

Line 243: Please note that there is no 2.11.2 Therefore no need to write 2.11.1 instead to keep it under one title one to save a consistent numbering list.

Line 246: Please add the sources of these two types of bacteria used in the study. Also, please add the catalog number for blood agar.

line 258: Please add the year for this version of Prism software.

Line 272: Please add the link of this server for quick access and explanation of the figure.

Line 291: please add the meaning/ indication of purple arrows to the figure legend.

Line 610: please add the meaning of X, XI and XII to the figure legend.

Line 709: Please mention the number of any internal project used to support this study with lab materials and other equipment.

References # 25, 48, 49 and 53 are incomplete. Please revise.

Comments on the Quality of English Language

Minor editing for English language is required.

Author Response

Please see the attachment. Thank you so much for your time in reviewing our manuscript. 

Reviewer 2 Report

Comments and Suggestions for Authors

1.        Please clearly state the novelty of the study in the abstract and throughout the text. Explain how this research is different from existing studies and how it advances antimicrobial drug discovery.

2.        The use of non-drug compounds like gallic acid, parabens, and shikimate does not match the claim that the study focuses on repurposed FDA-approved drugs. Please adjust this claim to reflect the actual focus of the study.

3.        In Figure 10, what do the arrows and angles mean? Please explain this clearly in the caption or text.

4.        Please provide a summary of the docking results for other compounds screened. Explain why these compounds were excluded from further analysis.

5.        Please clarify how the FMO analysis relates to the main topic of the study.

6.        In Figure 13, the y-axis uses a logarithmic scale, but this is not mentioned in the caption. Also, what statistical tests were done to confirm differences between the controls and the tested compounds? Please include this information.

7.        Please give more details about the MIC and MBC methods. Why were penicillin G and cefuroxime used as controls? The comparison of their results with X, XI, and XII is not discussed enough.

8.        Please add a section about the study's limitations and suggest future directions, such as in vivo testing or toxicity studies.

9.        Please compare the identified compounds with existing drugs to show their unique properties or advantages.

Author Response

(The authors gave the same response as above.)

Round 2

Reviewer 2 Report

Comments and Suggestions for Authors

I appreciate the authors' efforts in revising the manuscript, but I still have concerns about the antimicrobial activity assays described in Section 3.9. While MIC and MBC values are reported, there is no mention of the statistical methods used to analyze the data:

  • Were the results derived from multiple replicates? If so, how was variability assessed, and what statistical tests were applied to ensure reliability and reproducibility?
  • Was any statistical comparison conducted between the controls (penicillin G and cefuroxime) and the test compounds (X, XI, XII)?

The absence of statistical analysis undermines the validity of the reported results and weakens the authors' conclusions. Without this critical information, it is difficult to evaluate the robustness of the study, and the manuscript should not be published until this issue is adequately addressed.

Author Response

(The authors gave the same response as above.)

Round 3

Reviewer 2 Report

Comments and Suggestions for Authors

I have reviewed the revised manuscript and the authors' responses to the referees’ comments. The authors have sufficiently addressed the concerns raised in the original review. I believe the manuscript is now suitable for publication in Metabolites.